# Macrophages Cytokine Spp1 Increases Growth of Prostate Intraepithelial Neoplasia to Promote Prostate Tumor Progression

**DOI:** 10.3390/ijms23084247

**Published:** 2022-04-12

**Authors:** Justin K. Messex, Crystal J. Byrd, Mikalah U. Thomas, Geou-Yarh Liou

**Affiliations:** 1Center for Cancer Research and Therapeutic Development, Clark Atlanta University, Atlanta, GA 30314, USA; jmessex@cau.edu; 2Department of Biological Sciences, Clark Atlanta University, Atlanta, GA 30314, USA; crystal.byrd@students.cau.edu (C.J.B.); mikalah11@gmail.com (M.U.T.)

**Keywords:** macrophage, Spp1, osteopontin, prostate intraepithelial neoplasia, prostate tumor growth, integrin, CD44, 3D culture

## Abstract

Prostate cancer development and progression are associated with increased infiltrating macrophages. Prostate cancer is derived from prostatic intraepithelial neoplasia (PIN) lesions. However, the effects macrophages have on PIN progression remain unclear. Here, we showed that the recruited macrophages adjacent to PIN expressed M2 macrophage markers. In addition, high levels of Spp1 transcripts, also known as osteopontin, were identified in these macrophages. Extraneously added Spp1 accelerated PIN cell proliferation through activation of Akt and JNK in a 3D culture setting. We also showed that PIN cells expressed CD44, integrin αv, integrin β1, and integrin β3, all of which have been previously reported as receptors for Spp1. Finally, blockade of Akt and JNK activation through their specific inhibitor completely abolished macrophage Spp1-induced cell proliferation of PIN. Hence, our data revealed Spp1 as another macrophage cytokine/growth factor and its mediated mechanism to upregulate PIN cell growth, thus promoting prostate cancer development.

## 1. Introduction

Prostate cancer is the most common type of cancer in senior men. Fortunately, the 5-year survival rate of prostate cancer is approximately 100% due to diagnoses in early stages [1,2]. Although prostate cancer patients experience a high 5-year survival rate, prostate cancer is currently the second leading cause of cancer-related deaths [3] in men due to a lack of motivation to eliminate cancer through prostatectomy, which may lead to sexual dysfunction. Most patients with early stage prostate cancer prefer active surveillance and observation over proactive aggressive treatment choices, such as invasive surgeries. Due to the slow development typically seen in this form of cancer, many patients do not feel a need for urgency as they do with other forms of cancer. Unfortunately, diagnostic tools that can accurately monitor the transition from early stage to metastatic disease do not exist. Furthermore, there are no effective therapeutic measures to eradicate the castrate-resistant invasive cancer cells observed in the late stage of the disease, leading to the high death toll observed in prostate cancer patients.

Elevated inflammation has been associated with the aging process, and plays a crucial role in prostate cancer initiation, development, and progression [4,5,6,7]. Inflammation is known for its activation of macrophages, cells that play a critical role in maintaining a healthy immune response. Furthermore, these activated macrophages are known for their impact on various biological processes through the utilization of their secreted cytokines and chemokines [8,9]. Furthermore, environmental cues have also been shown to contribute to the phenotype expressed by these macrophages, classically activated (M1 inflammatory), or alternatively activated (M2 immunosuppressive) macrophages [10,11]. M1 inflammatory macrophages that are activated by lipopolysaccharide (LPS) and interferon gamma (IFNγ) possess high levels of reactive oxygen species (ROS) to eliminate various pathogens and unwanted cells. These macrophages typically express iNOS, CD38, tumor necrosis factor alpha (TNFα), and signal transducer and activator of transcription 1 (STAT1), all of which are used as M1 macrophage markers. On the other hand, M2 macrophages activated by interleukin-13 (IL-13) or IL-4 participate in wound healing and immunosuppression through their upregulation of IL-10, tumor growth factor β (TGFβ), and growth factors. These M2 macrophages express CD163, CD206, Ym1, and Arg 1, which are commonly used as M2 macrophage markers [12]. 

Several lines of evidence indicate that infiltrating macrophages, often referred to as tumor-associated macrophages (TAMs), are present in prostate cancer [13,14,15,16,17,18,19]. However, the correlation between TAMs and prostate cancer prognosis has remained debatable due to earlier reports, 2005 publications and prior, that have limited knowledge of the macrophage polarization process and specific markers distinguishing the various phenotypes [13,14,15,16,17,18,19]. Furthermore, severe limitations in technology utilizing the simultaneous detection of more than one antibody within the same tissue specimen have hindered our understanding of TAMs effect on prostate cancer. Fortunately, current technology has allowed us to analyze M1 and M2 macrophage markers in prostate cancer patient tissue samples, revealing an abundance of M2 macrophages existing within human prostate cancer tissues and that may predict a poor diagnosis [13,15,18]. 

Prostatic intraepithelial neoplasia (PIN) are pre-cancerous lesions of prostate cancer. The mechanisms of inflammation participating in PIN formation during prostate cancer development remain largely unknown. Herein, we report that infiltrating macrophages in human PIN lesions expressed the M2 macrophage marker CD163. In addition, conditioned media of PIN cells polarized macrophages to an M2 phenotype with high levels of Spp1 expression. When co-culturing with PIN cells in 3D, macrophages accelerated PIN cell growth. Moreover, this augmented effect of PIN cell proliferation can be achieved by exogenously added recombinant Spp1. We also showed that PIN cells expressed Spp1 receptors, including CD44, integrin αv, and integrin β1 and β3. Stimulation of PIN cells by macrophage-secreted Spp1 in 3D activated Akt and JNK, both of which led to cell proliferation of PIN cells. Overall, our data revealed a paracrine mechanism for macrophage-secreted Spp1 to expedite PIN cell proliferation through Spp1 receptors and subsequent activation of Akt and JNK. These results expand our understanding on the regulation of prostate cancer development, including its pre-cancerous lesion, which not only contributes to new therapeutic strategies but also provides insight into metastatic prostate cancer prevention. 

## 2. Results

### 2.1. Increased Infiltrating Macrophages in Human Prostate Intraepithelial Neoplasia Expressed M2 Macrophage Markers

We have previously shown that macrophages accelerated cell proliferation of prostate intraepithelial neoplasia under a 3D setting when they were co-cultured with PIN cells [7]. Furthermore, high numbers of infiltrated macrophages were detected in the areas of human PIN (Figure 1A, [1]). To gain further insight into the role of PIN-associated macrophages, we cultured Raw 264.7 macrophages in Pr111 PIN media, which led to the promotion of PIN cell proliferation [7] as compared to the control media. Using iNOS as a marker for M1 inflammatory macrophages [20], and Ym1 as a marker for M2 immunosuppressive macrophages [12], we found that the Raw 264.7 macrophages promoting PIN cell proliferation highly expressed Ym1 over iNOS (Figure 1B), suggesting that they are a M2 subtype. Our results were validated when using other reported markers for M1 inflammatory and M2 immunosuppressive macrophages, including CD38 [21,22] and CD206 [12] (Figure 1C). In addition, using CD68 as a pan-maker for human macrophages, a large portion of infiltrating macrophages in the PIN areas of human prostates were CD206 positive (Figure 1D). Altogether, these data suggest that M2 macrophages bolster prostate cancer development by increasing the cell growth of precancerous lesions, PIN.

### 2.2. Spp1 Highly Expressed in M2 Macrophages Induced Cell Proliferation without Influencing Apoptosis of PIN 

We next focused on identifying potential growth factors upregulated by M2 Raw 264.7 macrophages cultured in Pr111 PIN media using the RT2 Profiler PCR Array. As shown in Figure 2A, upregulated or downregulated genes of growth factors with at least a 2-fold increase or decrease were identified. The top-ranked upregulated growth factors include Spp1, macrophage-colony stimulating factor-1 (CSF-1), glial cell line-derived neurotrophic factor (GDNF), IL-6, IL-7, and IL-1α; the top-ranked downregulated growth factor includes insulin-like growth factor-1 (IGF-1). In addition, we also detected decreased gene expression of actin and HSP90. We selected the first three top-ranked growth factors: Spp1, CSF-1, and GDNF, for further confirmation of their effect on cell proliferation of murine Pr111 PIN cells in our 3D culture system. In order to evaluate cell proliferation in a 3D culture setting, nuclear cyclin D1 was used as a surrogate readout [4,7]. The addition of recombinant Spp1 significantly increased nuclear cyclin D1-positive Pr111 cell numbers as compared to the control (Figure 2B), suggesting that Spp1 promoted PIN cell proliferation. However, neither recombinant CSF-1 nor GDNF statistically enhanced Pr111 cell proliferation (Appendix A). In human prostate tissues, an increased expression of Spp1, known as osteopontin, was present not only in PIN cells but also in the stroma consisting of infiltrating macrophages (Appendix A). In addition to cell proliferation, we investigated whether macrophage-expressed Spp1 affects cell apoptosis of PIN. Using cleaved poly(AD-ribose) polymerase (PARP) as a marker for cell apoptosis, we found that while staurosporine, a well-reported apoptotic inducer [23,24], resulted in PARP cleavage, recombinant Spp1 treatment did not (Figure 2C). Furthermore, neither the Spp1-stimulated Pr111 PIN cells activated caspase 3, an essential protease for mediating apoptosis (Figure 2D). Altogether, these data indicated that macrophage-expressed Spp1 had no effect on cell apoptosis in a 3D culture setting.

### 2.3. PIN Cells Expressed Spp1 Receptors 

Previous reports have shown that Spp1, also known as osteopontin, transmits signals through binding to its receptors, including CD44 [25,26], and multiple integrins, such as αvβ3, αvβ1, αvβ5, α9β1, and α4β1 [27,28,29]. Therefore, it is very likely that macrophage-expressed Spp1 binds and signals through its receptors on the PIN cells, leading to increased PIN cell proliferation. To test this possibility, we first evaluated the expression of CD44, integrin αv, β1, and β3 in Pr111 PIN cells that were cultured in 3D. As shown in Figure 3A, regardless of Spp1 stimulation, Pr111 PIN cells indeed expressed all reported receptors for Spp1. Furthermore, the expression of CD44, integrin αv, and integrin β1 and β3 was also detected in PIN cells of human prostate tissues (Figure 3B–E). Altogether, these data suggested PIN cells expressed the receptors for Spp1 to mediate Spp1-transduced signals, resulting in cell proliferation of PIN cells.

### 2.4. M2 Macrophage Spp1 Activated Akt and JNK in PIN Cells

Mitogen-activated protein kinases (MAPKs) are well known for their role in regulating cell proliferation in response to growth factor stimuli [30]. In addition, PI-3 kinase and its downstream Akt signaling pathway also positively modulate cell growth [31,32]. To assess whether these protein kinases participate in Spp1-induced PIN cell proliferation, we utilized our 3D culture model, in which Pr111 PIN cells were treated with or without recombinant Spp1 followed by immunoblotting to evaluate the levels of activated Akt, extracellular signal-regulated kinase 1/2 (ERK1/2), c-Jun N-terminal kinase (JNK), and p38 MAPK (Figure 4A,B and Appendix A). We found increased levels of phosphorylated Akt and JNK in Spp1-treated Pr111 PIN cells in comparison to control cells, whereas the levels of activated ERK1/2 and phospho-p38 MAPK remained similar or slightly reduced in Pr111 PIN cells when treated with recombinant Spp1. Activation of Src and nuclear factor kappa B (NF-κB) has also been reported upon Spp1/osteopontin stimulation [33,34,35]. As shown in Figure 4B, Spp1 stimulation slightly decreased Src activation in Pr111 PIN cells. Nuclear factor of kappa light polypeptide gene enhancer in B cells inhibitor alpha (IκBα) inhibits NF-κB by masking the nuclear localization signals of NF-κB necessary to enter the cell nucleus, thus rendering NF-κB inactive and sequestering NF-κB in the cytoplasm compartment. Using the protein levels of IκBα as a surrogate readout for activation of NF-κB, we found that Spp1 did not activate NF-κB signaling in PIN cells (Figure 4C). In addition, we also detected increased levels of phosphorylated Akt in PIN cells of human tissue samples that had more infiltrating macrophages (Figure 4D). 

### 2.5. Inhibition of Akt and JNK Activation Reduced Macrophage Spp1-Induced PIN Cell Proliferation 

To test whether activated Akt and/or JNK signaling contributes to macrophage Spp1-induced PIN cell proliferation, we simultaneously added a JNK-specific inhibitor, SP600125, and/or an Akt-specific inhibitor, Wortmannin, to Pr111 PIN cells that were treated with macrophage Raw 264.7-conditioned media in a 3D setting. As shown in Figure 5A, treating PIN cells with either SP600125 or Wortmannin alone partially reduced Raw 264.7-conditioned media-induced cell proliferation of PIN. Furthermore, a dual treatment of both inhibitors had no additional blocking effect on PIN cell proliferation stimulated with macrophage-conditioned media as compared to each single treatment of SP600125 or Wortmannin. This suggested that the Akt signaling pathway cross talks with the JNK signaling pathway to regulate macrophage-induced PIN cell proliferation. Finally, when Pr111 PIN cells were stimulated with recombinant Spp1, inhibition of activated Akt and/or JNK by Wortmannin and/or SP600125 completely abolished Pr111 cell proliferation (Figure 5B), indicating that macrophage Spp1 induces PIN cell proliferation through activation of Akt and JNK signaling.

## 3. Discussion

The tumor microenvironment plays a pivotal role in cancer development, progression, and dissemination. Increased infiltration of activated macrophages and other types of immune cells has been observed in various types of cancer, including prostate cancer, to reshape the local environment, favoring cancer growth, migration, and invasion [36,37,38]. While many studies focused on the relationship between macrophages and prostate cancer, very little is known about how these immune cells affect prostate cancer precursors, known as prostate intraepithelial neoplasia (PIN). Our previous study demonstrated that PIN cells recruited macrophages through their expressed ICAM-1 and CCR2 [7]. Furthermore, macrophage-secreted factors, including C5a, CXCL1, and CCL2, increased cell proliferation of PIN [7]. In our current study, we identified Spp1, known as osteopontin, as another macrophage-secreted cytokine that promotes prostate tumor progression through upregulation of PIN cell growth without affecting cell death (Figure 2B,C). In addition to Spp1, several secreted factors, including IL-1α, IL-6, IL-7, and IGF-1, were also affected in the macrophages that promoted PIN cell proliferation in our screening through qRT-PCR arrays (Figure 2A). These factors require further verification as it pertains to their role in PIN cell proliferation. Of note, the expression of IGF-1 variants has been associated with prostate cancer development, including in human PIN tissues [39]. 

Osteopontin has been associated with various types of cancer invasion and metastasis, including prostate cancer [40,41,42,43,44]. It has been reported that an increased expression of osteopontin seems to be linked to prostate cancer development and progression, such as PIN and prostate cancer, and metastatic cancer cells in several transgenic mouse models of prostate cancer, including cRXRα^−/−^ and cPten^−/−^ mice [45]. In addition, human prostate cancer cell lines, including PC-3, DU145, LNCaP, and CWR22Rv1, indeed expressed osteopontin mRNA and proteins. Overexpression of osteopontin in LNCaP cells and PC-3 cells enhanced cell proliferation and cancer cell invasion ability. Due to these effects of osteopontin on prostate cancer cells, osteopontin expression levels have also been reported to be a prognostic marker for patient survival [46]. In human prostate tissues, there was no difference in osteopontin expression between normal prostate tissue and benign prostate hyperplasia (BPH). However, higher expression of osteopontin was greatly associated with malignant carcinoma tissues, increasing Gleason scores of the carcinoma, and reduced survival time of the prostate cancer patients. Our data (Appendix A) showed increased expression of osteopontin in the epithelial cells of PIN and upregulation in the stroma due to the secretion of osteopontin from nearby macrophages. 

We showed that PIN cells expressed osteopontin receptors, including CD44, integrin αv, and integrin β1 and β3 (Figure 3), suggesting a paracrine mechanism for PIN cell growth induced by macrophage-secreted Spp1/osteopontin (Figure 5C). Interestingly, in addition to its famous function as a cancer stem cell marker and the regulation in cell adhesion, CD44 has been shown to bind to osteopontin, leading to cancer cell movement, motility, and chemotactic behavior [25,26]. Loss of CD44 has been indicated in most prostate cancer due to extensive hypermethylation of CpG [47,48]. For example, the levels of CD44 expression increased in cells originating from high-grade prostate cancer compared to CD44 levels found in normal prostate epithelial cells. Furthermore, highly invasive human prostate cancer cell lines, including PC3 and TSU-Pr1, expressed increased levels of CD44 proteins compared to non-metastatic cancer cells. In total, 60% of primary prostate cancer tissues were moderately or strongly positive for CD44. Consistently, a majority of metastases of prostate cancer had non-detectable CD44. CD44 isoforms, including CD44H, CD44v6, or CD44v9, detected in humans [49,50] were linked to tumor differentiation status. Several integrin heterodimers, including αvβ3, αvβ1, αvβ5, αvβ6, α9β1, α5β1, and α4β1 [27,28,29,50,51,52,53,54], have been demonstrated to be osteopontin receptors. Furthermore, the interaction between osteopontin and CD44 variants can be abolished by the antibody directed against integrinβ1 [25]. Although, so far, little is known about osteopontin and integrin heterodimers in PIN during prostate tumor development, in prostate cancer, the integrin heterodimer αvβ3 has been identified [55]. 

We showed that inhibition of both activated Akt and JNK using Wortmannin and SP600125 blocked macrophages and its secreted Spp1-induced PIN cell proliferation (Figure 5A,B). This data suggests an anti-tumor role of Wortmannin and SP600125 in prostate tumor progression. Due to the challenges of low solubility in the water-like solvent of Wortmannin, other Akt inhibitors being used in clinical trials for prostate cancer include capivasertib, an oral pan-Akt inhibitor, and ipatasrtib, another oral small compound that binds to the ATP-binding site of Akt [56,57]. Interestingly, a few studies have reported a revival of Wortmannin in prostate cancer therapy [58,59]. These include the use of a nanoparticle drug delivery system to overcome the solubility and cytotoxicity of Wortmannin and an increase in its mediated therapeutic efficacy. In addition to Akt activation, activated JNK signaling has also been well reported [60]. Despite many studies reporting on the efficiency of JNK inhibition as an option to suppress prostate cancer progression and metastasis using the inhibitor SP600125, the use of JNK inhibitors as a clinical trial therapeutic option is premature [61]. 

In summary, our data provided a mechanistic link between M2 macrophages and increased prostate tumor progression through macrophage-secreted Spp1/osteopontin to enhance cell growth of prostate pre-cancerous lesions PIN. Activation of osteopontin receptors expressed on PIN cells, such as CD44 and integrin αv, β1, and β3, subsequently activated Akt and JNK pathways, leading to higher tumor cell proliferation. Small molecular compounds that can specially target any molecules presented in this mechanism (Figure 5C) to disrupt prostate tumor progression may be used as new strategies to prevent from or reduce death of metastatic prostate cancer. 

## 4. Materials and Methods

### 4.1. Cell Lines, Antibodies, and Reagents 

Murine macrophage Raw 264.7 cells (ATCC; Manassas, VA, USA) were cultured in RPMI-1640 containing 100 U/mL penicillin/streptomycin and 10% FBS. Murine PIN Pr111 cells were cultured on collagen I-coated dishes and maintained as described previously [7,62]. In brief, Pr111 cells were cultured in PrEBM^TM^ prostate epithelial cell growth basal media containing MEGM singlequots, 2% FBS, 1 mM sodium Pyruvate, 10 nM dihydro-testosterone, and 100 U/mL penicillin/streptomycin. Cells were maintained in a 37 °C incubator with 5% CO_2_. The Pr111 cells used in all experiments were less than 10 passages. The Raw 264.7-conditioned media was obtained from 5 × 10^5^ cells per well of a 6-well dish cultured with Pr111 complete media for 2 days. Conditioned media was freshly prepared for each experiment. Antibodies of iNOS, CD38, CD206, CD68, and CD163 were purchased from Abcam (Cambridge, UK). Antibodies containing cyclin D1, PARP, phospho-Akt, Akt, phospho-ERK1/2, ERK1/2, phospho-JNK, JNK, phospho-p38 MAPPK, p38 MAPK, phospho-Src, Src, and IκBα were purchased from Cell Signaling Technology (Danvers, MA, USA). F4/80 antibody was obtained from Invitrogen (Waltham, MA, USA). Both integrin αvβ3 and CD44 antibodies were obtained from R & D systems (Minneapolis, MN, USA). Ym1 antibody was obtained from Stemcell Technologies (Vancouver, Canada). In addition, all antibodies are listed in Appendix A. Mouse recombinant Spp1 was obtained from BioLegend (San Diego, CA, USA). Recombinant CSF1 and GDNF were purchased from PeproTech (Rocky Hill, NJ, USA). Matrigel was obtained from Corning (Corning, NY, USA). Wortmannin and SP600125 were obtained from Cell Signaling Technology and Tocris Bioscience (Bristol, UK), respectively. The other reagents used are described in the specific experiment sections. 

### 4.2. 3D Culture of Pr111 Cells and Treatment

Pr111 single cells were plated on top of 50% Matrigel that was mixed in an equal volume of Matrigel and Pr111 complete media. The cells were stimulated with the indicated recombinant proteins and indicated concentrations (recombinant Spp1: 250 ng/mL; recombinant CSF-1: 100 ng/mL; recombinant GDNF: 50 ng/mL) or vehicle control (ddH_2_O). The used concentration, especially for recombinant Spp1, was determined according to the results from dose curve experiments for the minimal dose required in Pr111 cells and the reported minimal required dosage in the literature. For the treatment, 0.5 µM Wortmannin, 1 µM SP600125, or vehicle DMSO was added to the cells simultaneously with or without recombinant Spp1 or Raw 264.7-conditioned media after cells were seeded on Matrigel. 

### 4.3. Cell Proliferation Assay of Pr111 Cells Cultured in 3D

Pr111 cells were grown in 3D Matrigel culture with the indicated stimulation/treatment for 72 h (when macrophages were recruited by Pr111 cells [7]) and were rinsed twice with PBS (140 mM NaCl, 2.7 mM KCl, 8 mM Na_2_HPO_4_, and 1.5 mM KH_2_PO_4_ (pH 7.2)), fixed with 4% paraformaldehyde for 20 min, permeabilized with half percent TritonX-100/PBS for 20 min, 0.1% SDS/PBS for 1 min, and blocked in 10% goat serum for 1 h at room temperature. Samples were incubated with anti-cyclin D1 antibody (1:250) overnight at 4 °C followed by 3 washes of PBS containing 0.05% Tween-20. Secondary antibody goat-anti-rabbit conjugated with Alexa Fluor 488 (Thermo Fisher Scientific) at a dilution of 1:500 was added to samples for 30 min at room temperature. After another three washes of PBST, cell nuclei were labeled with DAPI. Images were collected in ibidi mounting media. Serial sections of images (Z-stack) were captured by a Zeiss Axiovert 200M inverted fluorescent microscope using a 10× objective lens. Each 3D image was reconstructed from Z-stack images. Numbers of nuclear cyclin D1 were counted randomly, and each condition contained five 3D images. The cell proliferation index was calculated using the ratio: numbers of nuclear cyclin D1 divided by number of all cells (DAPI). 

### 4.4. Fluorescent Microscope Imaging

Raw 264.7 macrophages cultured in either Pr111 complete media or control media for 24 h were fixed in 4% paraformaldehyde for 20 min, permeabilized with 0.5% Triton X-100 in PBS for 10 min, 0.1% SDS in PBS for 1 min, and blocked in 10% goat serum for 1 h at room temperature. F4/80 antibody (1:200, Invitrogen) was used to label all macrophages. Both iNOS (1:250, Abcam) and CD38 (1:200, Novus Biologicals) were used to label M1 macrophages. Both Ym1 (1:200, Stemcell Technologies) and CD206 (1:1000, Abcam) were used to label M2 macrophages. After 3 washes with PBS containing 0.05% Tween-20, samples were incubated with appropriate Alexa Fluor-conjugated secondary antibodies and DAPI. After 3 washes with PBST, images were captured using a Zeiss Axiovert 200 M inverted fluorescent microscope with a 10× or 20× objective lens. 

### 4.5. Growth Factor Real-Time qPCR Array

The growth factors expressed by Raw 264.7 macrophages cultured in Pr111 completed media or control media were examined using the RT2 Profiler PCR Array with catalog number PAMM-041Z (Qiagen (Hilden, Germany)) according to the manufacturer’s instructions. In brief, RNA was isolated from Raw 264.7 macrophages that were cultured in either Pr111 completed media or control media using the RNeasy plus mini kit (Qiagen). Complementary cDNA was converted using the RT First Stand Kit (Qiagen) followed by real-time qPCR array for growth factors. All necessary procedures during the qPCR array were carried out based on the manufacturer’s manual. 

### 4.6. Cell Lysates Collection and Immunoblotting

Cells were rinsed twice with cold PBS, lysed in buffer A (50 mM Tris/HCl (pH 7.4), 1% TritonX-100, 5 mM EDTA (pH 7.4), and 150 mM NaCl) or RIPA buffer (25 mM Tris/HCl (pH 7.5), 1% Triton X-100, 140 mM NaCl, 1 mM EDTA, 0.5% SDS) containing protease inhibitor cocktail (Thermo Fisher Scientific), vortexed at the maximal speed for 1 min, incubated at 4 °C for 10 min, and centrifuged at the same temperature, at 14,000 rpm for 10 min. The supernatants were denatured followed by SDS-PAGE. Resolved proteins were transferred to nitrocellulose membranes prior to blocking in 5% BSA in TBST (50 mM Tris.HCl (pH 7.6), 0.05% Tween 20, 150 mM NaCl), and incubated with the antibodies of interest in 5% BSA in TBST until the next day at 4 °C. The appropriate horseradish peroxidase-conjugated secondary antibodies were added to the membranes for 30 min at room temperature. Images were captured and visualized with ECL and X-ray film.

### 4.7. Immunohistochemistry 

Human prostate tissue slides, including normal or cancer tissue samples, were obtained from BioChain (Newark, CA, USA), US Biomax (Derwood, MD, USA), and Cooperative Human Tissue Network (CHTN). In brief, slides were deparaffinized in xylene and gradually re-hydrated through 100% alcohol to distilled water. The rehydrated slides were subjected to heat-induced antigen retrieval in the antigen retrieval buffer, including either 10 mM sodium citrate buffer (pH 6.0) or 10 mM Tris, 1 mM EDTA buffer (pH 9.0). Slides were incubated with 3% hydrogen peroxide to decrease endogenous peroxidase activity and rinsed with PBS. Slides were treated with protein block serum-free solution (DAKO) for 10 min at room temperature. After the primary antibody (CD68 1:500 (DAKO); CD44 1:200 (BioLegend); integrin αv 1:500 (ProteinTech); integrin β1 1:1000 (ProteinTech); integrin β3 1:750 (Thermo Fisher Scientific); phospho-Akt 1:75 (Cell Signaling Technology)) was incubated with the slides, the ImmPRESS Polymer Detection Kit (Vector Laboratories (Burlingame, CA, USA)) was used according to the manufacturer’s instructions to visualize the stained slides. For M2 macrophages in the human prostate intraepithelial neoplasia areas, primary antibodies, including CD68 from DAKO at a dilution of 1:500 and CD206 from Abcam at a dilution of 1:1000, were incubated with the tissue samples as described above. Appropriate secondary antibody conjugated with Alexa Fluor 488 and Alexa Fluor 594 were applied to tissue samples for 30 min at room temperature followed by DAPI staining to label nuclei. Images were collected using the Aperio VERSA tissue scanner with ImageScope software (Aperio (Sausalito, CA, USA)). 

### 4.8. Statistical Analysis

All results shown represent 3 independent experiments or tissue samples. Data are presented as ±SE while *p*-values were acquired using the Student’s *t* test to compare 2 sets of data using Prism (GraphPad Software (San Diego, CA, USA)). For data sets with more than 2 sets of data, 1-way ANOVA analysis along with multiple comparisons was carried out using Prism (GradPad Software). *p* < 0.05 was considered statistically significant.

## Figures and Tables

**Figure 1 ijms-23-04247-f001:**
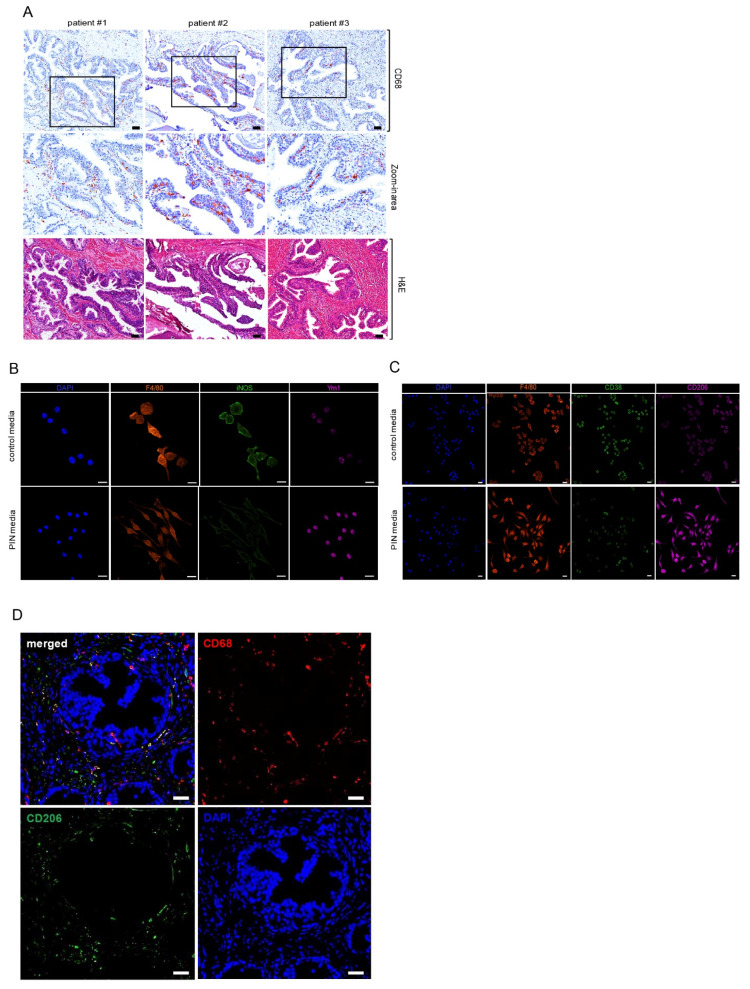
Increased infiltrating macrophages in the prostate intraepithelial neoplasia were M2 subtype. (**A**) Human tissue samples of prostate cancer containing the area of prostate intraepithelial neoplasia (PIN) were immunostained with the antibody of CD68, a pan marker for macrophages (top panel, and middle panel: zoom-in areas). Hematoxylin and eosin (H & E) staining was carried out to visualize the morphology of the prostate cancer tissues (bottom panel). Scale bar: 25 µm. (**B**,**C**) Raw 264.7 macrophages cultured with either control media or Pr111 PIN complete media were fixed, permeabilized, and immunostained with antibodies for markers of pan macrophages, F4/80; M1 macrophages including iNOS (**B**) and CD38 (**C**); M2 macrophages including Ym1 (**B**) and CD206 (**C**). Scale bar: 20 µm. (**D**) Human tissue samples of PIN were immunostained with antibodies of CD68 and CD206 for the assessment of all macrophages (green) and M2 macrophages (red), respectively. Nuclei were labeled by DAPI and visualized. Scale bar: 50 µm. The image shown here is representative of at least 3 independent experiments or tissue samples.

**Figure 2 ijms-23-04247-f002:**
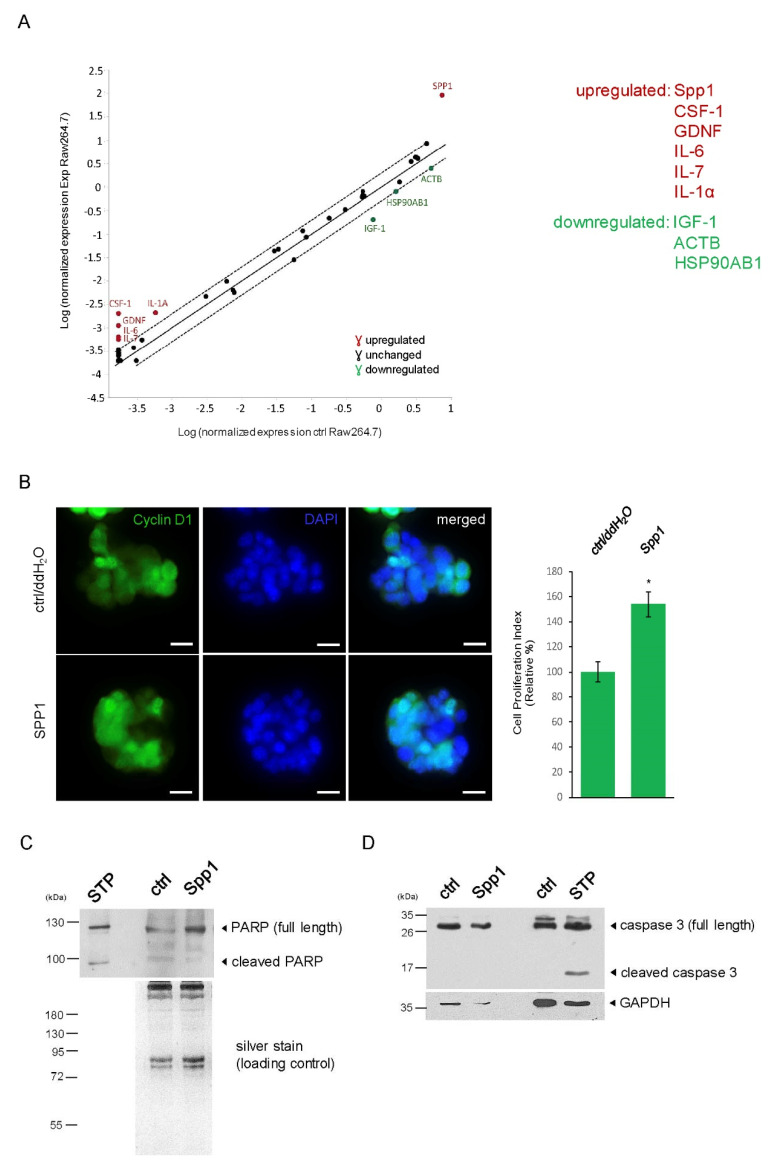
Macrophage Spp1 enhanced PIN cell proliferation without affecting apoptosis. (**A**) A plot demonstrating the identified genes that were more than 2-fold upregulated (red) or downregulated (green) in Raw 264.7 macrophages cultured in Pr111 PIN media (Y-axis, indicated as Exp Raw264.7) as compared to those in control media (X-axis, indicated as ctrl Raw264.7). (**B**) Murine PIN Pr111 cells cultured on Matrigel in 3D were treated with either recombinant Spp1 or control/ddH_2_O. At the endpoint, the cells were fixed, permeabilized, and immunostained with cyclin D1 (green), a surrogate marker for cell proliferation. Cell nuclei were stained by DAPI (blue). Cell proliferation index of Pr111 cells under these two conditions was quantified. Scale bar: 20 µm, *: *p* < 0.05 as compared to control. (**C**) The cell lysates isolated from (**B**) were subjected to immunoblotting for assessment of the PARP cleavage, an indication of apoptosis. Staurosporin (STP)-treated cell lysates were used as a positive control for PARP cleavage. (**D**) Similar to (**C**), the isolated total cell lysates were examined for cleaved caspase 3, an essential protease activated during apoptosis, using immunoblotting. STP-treated cell lysates serve as a positive control for cleaved caspase 3. The image shown here is representative of at least 3 independent experiments or tissue samples.

**Figure 3 ijms-23-04247-f003:**
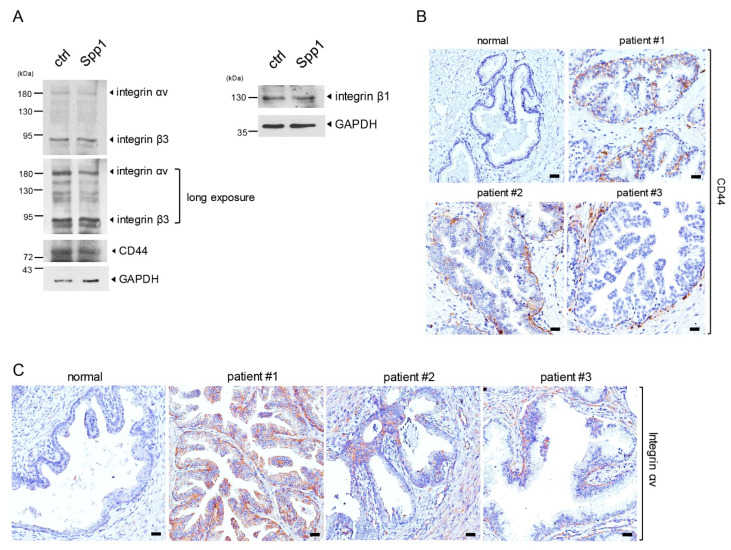
Expression of Spp1 receptors in PIN cells. (**A**) Cell lysates from murine PIN Pr111 cells grown on Matrigel in 3D treated with recombinant Spp1 or control for 72 h were subjected to immunoblotting for assessment of the Spp1 receptors, including integrin αvβ3, integrin β1, and CD44. Long exposure: longer time to catch the emitted signal from the WB membrane on the X-ray film. (**B**–**E**) Human tissue samples of normal prostate and prostate cancer containing PIN areas were immunostained for the expression of CD44 (**B**), integrin αv (**C**), integrin β1 (**D**), and integrin β3 (**E**). Scale bar: 50 µm. the image shown here is representative of at least 3 independent experiments or tissue samples.

**Figure 4 ijms-23-04247-f004:**
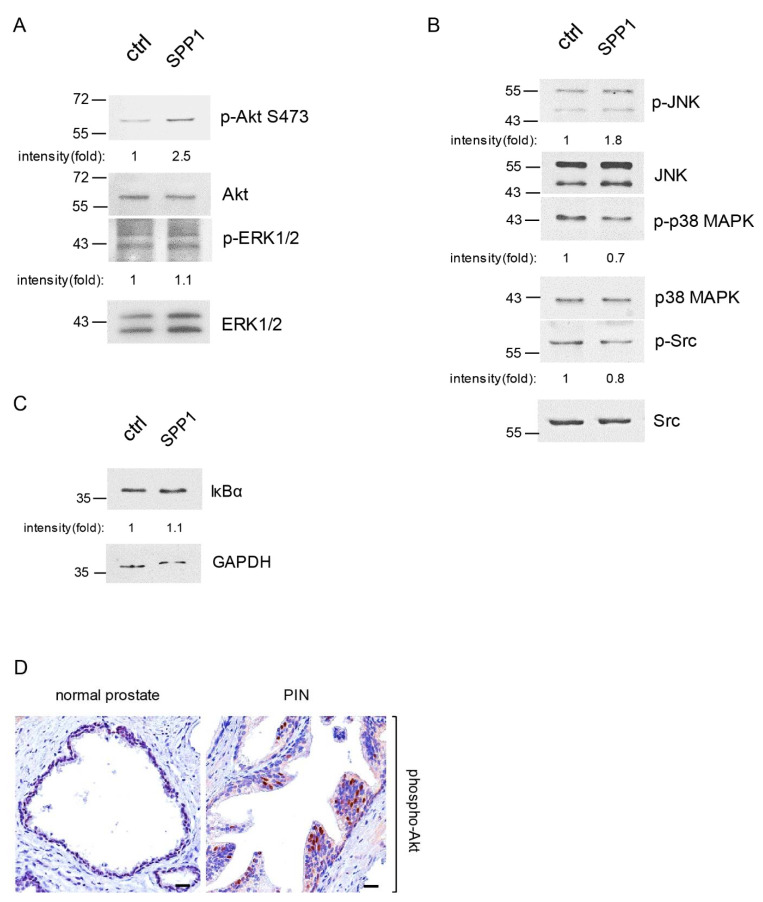
Macrophage Spp1 activated Akt and JNK signaling in PIN cells. (**A**–**C**) Pr111 cells cultured on Matrigel in 3D were stimulated with either control/ddH_2_O or recombinant Spp1 for 72 h. Cell lysates were collected and subjected to immunoblotting to examine the proteins of interest, including p-Akt, Akt, p-ERK, and ERK (**A**); p-JNK, JNK, p38 MAPK, p38 MAPK, p-Src, and Src (**B**); and IκBα (**C**). The relative intensity fold change was calculated by the ratio of phosphorylated protein levels over the corresponding total protein levels and set as 1 for the control in each case. Each image shown is representative of 3–5 independent experiments. (**D**) Human tissue samples of normal prostate and prostate cancer containing PIN areas were immunostained for phospho-Akt expression. Scale bar: 25 µm. *n* = 3.

**Figure 5 ijms-23-04247-f005:**
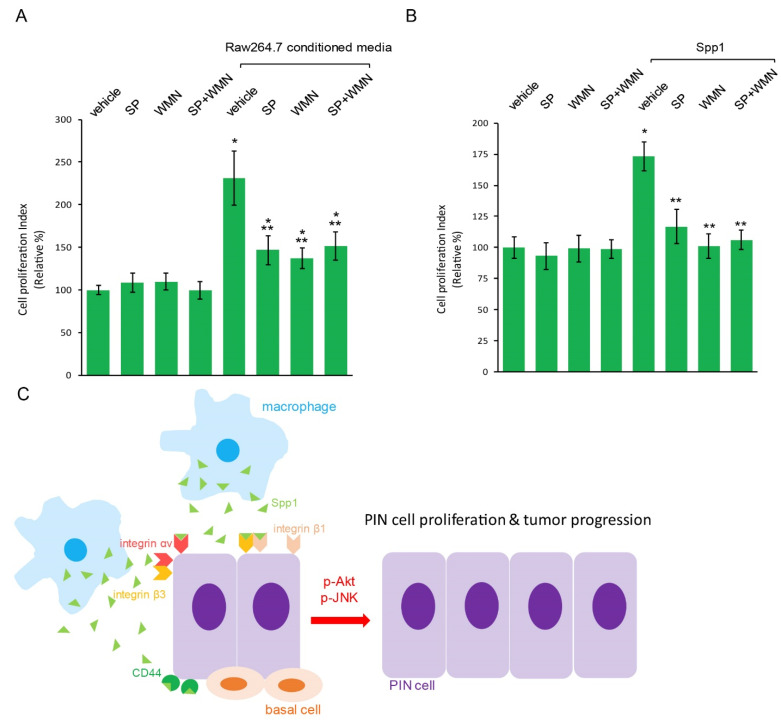
Blockade of Akt and JNK suppressed macrophage Spp1-induced cell proliferation of PIN. (**A**,**B**) SP600125, a specific JNK inhibitor, and/or Wortmannin, a specific Akt inhibitor, was added to 3D cultures of murine PIN Pr111 cells stimulated with either Raw 264.7-conditioned media (**A**) or recombinant Spp1 (**B**) for 72 h. Vehicle/DMSO was used as a control for SP600125 and Wortmannin; ddH_2_O was used as a control for recombinant Spp1. The cell proliferation index under these conditions was quantified. *: *p* < 0.05 as compared to the control+vehicle, **: *p* < 0.05 as compared to vehicle+Raw264.7-conditioned media or Spp1 (-way ANOVA and multiple comparisons), *n* = 3. Abbreviation: SP: SP600125; WMN: Wortmannin (**C**) A summarized scheme from our findings in this manuscript on how macrophage cytokine Spp1 accelerated prostate tumor progression via upregulation of cell growth of PIN, a precursor for prostate cancer.

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
