# Peer review of "Macrophages Cytokine Spp1 Increases Growth of Prostate Intraepithelial Neoplasia to Promote Prostate Tumor Progression"

_ijms, 2022, doi:10.3390/ijms23084247_

Round 1

Reviewer 1 Report

This paper reported the osteopontin-dependent upregulation of the prostate intraepithelial neoplasia. In my opinion, the manuscript is interesting and well written however, I found a few stumbles and doubts which need to be clarified in the text.
Below are provided the specific comments and suggestions.

Introduction - The information presented in the first paragraph is an overview of the prostate cancer statistics and treatment methods, however, there is a complete lack of references supporting the information presented in this fragment.
 Moreover, there is a lack of hypothesis and aim of the study. The last paragraph of the introduction is more conclusion or the brief of the results than the Introduction. Please reconsider.
Figure 1 D -  should be indicated in which tissues structure the specific markers for M1 and M2 macrophages were mainly found. Figure 1B, C, - it is shame the Authors did not merge DAPI and other macrophage-marker pictures. They may also indicate in which area of ​​the cells the markers are visible (nucleus or cytoplasm area).
Due to the Authors using the RT2 Profiler PCR Array the short information regarding this array should appear in the M&M section, and the Cat No. of the array must be necessary indicated.
Figure 2A and lines 116-118 - Authors indicated that top up-regulated genes were Spp-1, CSF-1, GDNF, IL-6, and IL-7, but on Fig. 2A there is also IL-1α up-regulated. Why Authors did not also focus on this cytokine? Please explain.
FIG. 2 for clarification –The X-axis should contain the description “control media”, whereas Y-axis - “Pr111media”.

The very interesting result is the down-regulation of ACTB, which is commonly used in many Real Time experiments as a housekeeping gene. Therefore, detailed information regarding the calculation of the gene expression changes should be provided. I am aware that the Authors focused on the Spp1, but in my opinion, a few sentences regarding the down-regulated genes as a result of the present study should appear in the discussion section.
LINES 149-150- this sentence must be rewritten because CD44 is not a ligand of SPP1, but its receptor (Weber et al. 1996).
Figure 3A, what does it mean for long exposure? There is neither information in the M&M section nor the results. This longer exposure is regarding the different time incubation with antibodies or with SPP1?
 Lines 173-184 – Authors mentioned “We found increased levels of phosphorylated Akt and JNK in Spp1-treated Pr111 PIN cells in comparison to control cells, whereas levels of activated ERK1/2  and phospho-p38 MAPK remained similar or slightly reduced in Pr111 PIN cells when treated with recombinant Spp1…….” but, besides the fold change, I did not find the statistical analyses.  Were the described changes statistically significant?
Figure 5 - to clarify the used abbreviations must be present in the Figure legend. 

There is no explanation why the Authors decided to select specific doses of treatments (especially Spp1) and the time of incubation with the treatments. The reason for the choice must be provided in the M&M section. 
How did the Authors isolate RNA and check the quality of obtained nucleic acid?  Did the RIN were obtained? Details regarding the qPCR array (condition of the reaction) must be presented.
It would be a great advance if the information concerning specific antibodies’ appeared in fragments regarding proliferation, immunoblotting of the M&M section. Moreover,  the Cat No. of antibodies used in the study should be provided.
There are typos and stylish mistakes. The abbreviations must be explained when for the first time appeared in the text.

Author Response

We thank the reviewer's suggestions and comments, and provide a point-to-point response (please see the attachment). Thank you.

Reviewer 2 Report

This manuscript describes the effects of the cytokine Spp1 on prostate tumor progression. The paper is well written and the results are presented accurately. The provided results are of interest and shed new light on the role of Spp1 for the promotion of prostate cancer. I recommend acceptance after minor revision:

Please correct the following quotations: Line 30: ´´… may lead sexual dysfunction.´´. Line 90: ´´… as a maker …´´? Line 244: ´´osteopotin´´.

Figure 2B: The resolution of this figure is not good, maybe due to the pdf transformation process. Please check again and improve the resolution if necessary.

Discussion: The effects of wortmannin and SP600125 should be discussed. How far can currently applied anticancer drugs help to suppress Spp1 and its tumor promoting effects? Therapy options should be outlined in a more detailed way.

Author Response

Comment 1: This manuscript describes the effects of the cytokine Spp1 on prostate tumor progression. The paper is well written and the results are presented accurately. The provided results are of interest and shed new light on the role of Spp1 for the promotion of prostate cancer. I recommend acceptance after minor revision.

Response: We thank the reviewer for the kind words and greatly appreciate the reviewers high praise for our manuscript.  

Comment 2: Please correct the following quotations: Line 30: ….may lead sexual dysfunction.”. Line 90: “….as a maker….”? Line 244:” osteopotin”.

Response: We thank the reviewer for carefully reading our manuscript here and we do apologize for these typos. Now these are corrected to be Line 30: “…..may lead to sexual dysfunction.”. Line 90: “….as a marker….”? Line 244:” osteopontin”

Comment 3: Figure 2B: The resolution of this figure is not good, maybe due to the pdf transformation process. Please check again and improve the resolution if necessary.

Response: We thank the reviewer for looking at our results carefully and do appreciate this suggestion.  We now enlarged this figure 2B and made it easier to our readers to see.

Comment 4: Discussion: The effects of wortmannin and SP600125 should be discussed. How far can currently applied anticancer drugs help to suppress Spp1 and its tumor promoting effects? Therapy options should be outlined in a more detailed way. 

Response: We thank the reviewer for this suggestion. We have followed this suggestion and have added another paragraph at the end of the discussion to briefly mention the possibility for using Wortmannin and SP600125 in prostate cancer therapy.

Round 2

Reviewer 1 Report

As I wrote previously, the manuscript is interesting and indicates new knowledge concerning Spp1-dependent PIN cell growth and the promotion of prostate cancer development. The Authors provided sufficient responses to all my queries. The present version of the manuscript is improved according to reviewers’ suggestions. It is a well-made paper. I have only a few minor comments presented below.
Respecting the response concerning the detailed data presented in the first paragraph of the introduction, this was not a critique, only a recommendation. However, I still believe that the information regarding the survival rate of prostate cancer and the cause of cancer-related deaths need to be supported by appropriate scientific references.
The information presented in the Reviewer's response concerning IL1a should be indicted in the short form in the results section. Similarly, the explanation of the longer exposure should be indicated in the Figure 3 legend.
In my opinion, besides osteopontin, the interesting result is the down-regulation of IGF1 and its possible role in prostatic intraepithelial neoplasia (PIN), please see Int. J. Mol. Sci. 2020, 21(19), 6995; https://doi.org/10.3390/ijms21196995. One to two sentences about the further direction of the Author's study (maybe they consider the interactions between SPP and IGF) should be provided in the Discussion.

Author Response

We thank the Reviewer's suggestions. Please see the provided point-to-point response in this 2nd round revision (see the attachment). Thank you.
